# Characterisation and localisation of plant metabolites involved in pharmacophagy in the turnip sawfly

**Leon Brueggemann, Lisa Johanna Tewes, Caroline Müller** [ORCID] *

Department of Chemical Ecology, Bielefeld University, Bielefeld, Germany

* caroline.mueller@uni-bielefeld.de

## Abstract

Several herbivorous insects consume certain metabolites from plants for other purposes than nutrition, such as defence. Adults of the turnip sawfly, *Athalia rosae* take up specific terpenoids, called clerodanoids, from *Ajuga reptans*. These metabolites are slightly modified by the sawflies and influence their mating behaviour and defence against predators. We characterised these metabolites and investigated their localisation in the insect and the specificity of the uptake and metabolite modification. Therefore, we performed feeding assays with adults and larvae of *A. rosae* as well as larvae of *Spodoptera exigua*, followed by chemical analyses. Two main clerodanoid-derived metabolites were detected in the abdomen and thorax but also on the surface of the adults. Small amounts were also found in larvae of the sawfly, while they were not detectable in *S. exigua*. Our findings provide new insights into the peculiarities of pharmacophagy and specialised metabolism in *A. rosae*.

## Introduction

Insects play an irreplaceable role in diverse ecosystems around the world [1] and half of them are herbivorous [2]. Plants, even more irreplaceable, use various metabolites that are specific to certain taxa as effective defence against such herbivores [3]. Thus, herbivores that feed on this plant material must be able to cope with such plant metabolites [4, 5]. One possible way is making use of these compounds by sequestration [6, 7], i.e. uptake and concentration of the plant metabolites directly or after modification [8]. However, for many plant metabolites taken up by herbivorous insects, little is known about the site of storage in the insect, the metabolite identity in case the plant compounds are further metabolised and the specificity of this metabolism.

An uptake of plant metabolites that serves other purposes than energy uptake is also called pharmacophagy [9]. Acquired substances can have different benefits ranging from an enhanced defence against predators to more frequent mating. A classic example are butterflies of the genus *Danaus* (Lepidoptera: Danaidae), which take up alkaloids from different host plants, such as *Eupatorium* spp. (Asteraceae), as adults. These compounds are converted to male courtship pheromones and serve to attract females [10]. Additionally, the butterflies are better protected

**Data Availability Statement:** Spectra and bucket tables will be made publicly available via Metabolights with identifier MTBLS7508.

**Funding:** This research was funded by the German Research Foundation (DFG) as part of the SFB TRR

 

212 (NC³), project no. 396777467 (granted to C. M.) The funders had no role in study design, data collection and analysis, decision to publish, or preparation of the manuscript.

**Competing interests:** The authors have declared that no competing interests exist.

against predators, such as mantids [11]. The range of plant and invertebrate species involved in sequestration and pharmacophagy is very broad and still demands research [6].

Pharmacophagy is also known in the turnip sawfly, *Athalia rosae* (Hymenoptera: Tenthredinidae). Larvae of *A. rosae* use exclusively Brassicacae species as host plants for feeding. They sequester glucosinolates from these plants which can help as protection against various predators such as ants and wasps [12, 13]. Adults take up nectar of Apiaceae [14], but have also been observed to "nibble" on plants of the genera *Clerodendrum* (Verbenaceae) and *Ajuga* (Lamiaceae), including bugleweed, *Ajuga reptans* [15]. This nibbling behaviour leads to an uptake of terpenoids, more specifically clerodane diterpenoids, called clerodanoids. When having acquired clerodanoids, the adults become sexually more attractive for their mating partners [16]. In addition, they are capable of obtaining the compounds through nibbling on the surface of conspecifics that had prior access to clerodanoids [17], indicating that the metabolites must be located at the surface. Furthermore, we observed that adult *A. rosae* clean themselves after contact with *A. reptans* leaves and may thereby spread some compounds over their body. Clerodanoid uptake by adults also influences the defence against antagonists, such as mantids [18] or potential pathogenic fungi [19]. Thus, acquisition of plant metabolites clearly changes the individual niche [20].

Clerodanoids are a diverse compound group that share a common bicyclic structure composed of a 5-membered ring fused with a 7-membered ring. About 25% of the clerodanoids have a 5:10 cis-ring bond. The remaining 75% of the clerodanoids have a 5:10 trans-bond. Clerodanoids with a 5:10 trans-bond are characteristic for Lamiaceae. They have been found to possess various biological activities such as anti-inflammatory, antitumor, antimicrobial, antiviral and insecticidal properties [21, 22]. The structure of the metabolites revealed in adult *A. rosae* does not exactly match the structure of clerodanoids found in *A. reptans* plant material and therefore the clerodanoids must be further modified by the insects [23]. When having nibbled on *A. reptans*, we found clerodanoid structures with masses of 482 and 484 in the adults [18], but the compounds have not been identified yet. Moreover, it was unclear where these compounds are stored in the body and whether this metabolism is specific to adults of *A. rosae*.

To characterise the compounds taken up from *A. reptans* by adults of *A. rosae*, we conducted feeding assays and metabolomics analyses via ultra high performance liquid chromatography coupled with quadrupole time-of-flight mass spectrometry (UHPLC-QToF-MS/MS). Using metabolomics is a powerful approach to compare the chemical differences in the metabolomes of organisms exposed to different treatments [24, 25]. Due to the size and the specific fragmentation pattern of clerodanoids, we expected chromatography combined with mass spectrometry to be a promising tool to investigate the chemical structures of undescribed molecules [26]. Different body parts and body washings of unexposed and clerodanoid-exposed adults of *A. rosae* were analysed to study the localisation of the sequestered compounds. Moreover, to explore the specificity of clerodanoid metabolism, we studied whether the metabolites with the masses 482 and 484 are likewise present in larvae of *A. rosae* as well as a herbivorous generalist, *Spodoptera exigua* (Lepidoptera: Noctuidae), using feeding assays combined with UHPLC-QToF-MS/MS analyses.

## Material and methods

### Rearing of insects and plants

Adult individuals of *A. rosae* were collected over the summer in the surroundings of Bielefeld, Germany. For a continuous rearing, adults were offered *Sinapis alba* (Brassicaceae) plants for oviposition and larvae were kept on non-flowering plants of *Brassica rapa* var. *pekinensis* (Brassicaceae) for feeding. Insects were kept in cages at a 16:8 h day:night cycle. Larvae

pupated in soil provided in Petri dishes or the potting soil. Emerging adult individuals were separated by sex, provided with 2% honey water and kept at ~5˚C in the dark until they were used for experiments. Plants of *A. reptans* were grown from seeds (RHS, London, United Kingdom) in a greenhouse and kept in a 16:8 h day:night cycle without specific climate control. Additional leaf material was collected around Bielefeld, Germany, and stored at -80˚C. Eggs of *S. exigua* were kindly provided by A. Steppuhn, University of Hohenheim. Larvae were reared on *B. rapa* var. *pekinensis* until the fourth instar and then used for feeding trials.

## Exposure trials with insects

Adult females (1–2 days after adult emergence) were placed individually in Petri dishes (5.5 cm diameter) lined with moist filter paper and offered fresh middle-aged leaf discs (6 mm diameter) of *A. reptans* plants for 1 h. During the exposure, the nibbling duration was tracked visually. After the exposure time, adults were immediately frozen at -80˚C and stored until sample preparation. These adults are labelled as "C+" in the following. Naive adults (without contact to leaves, labelled "C-") were frozen in parallel.

To investigate the specificity and potential metabolism of clerodanoids in larvae of *A. rosae* and a generalist species, feeding experiments were carried out with extracts of *A. reptans* leaves. For extract preparation, young leaves of flowering *A. reptans* plants were harvested, weighed, frozen in liquid nitrogen, lyophilised, weighed again and homogenised. The dried material was extracted twice for 10 min in ethyl acetate (LC-MS grade, VWR, Leuven, Belgium) in a concentration of 80 mg dw ml$^{-1}$ using an ultrasonic bath. Five technical replicates were prepared in parallel and aliquots were frozen until use. Fourth-instar larvae of *A. rosae* and *S. exigua* were starved for 2 h prior the trial. Since both species do not feed as larvae on *A. reptans*, 15 µl of a leaf extract of *A. reptans* (see below) were applied on leaf discs (6 mm diameter) of the host plant *B. rapa* var. *pekinensis* and offered on moistened filter paper to individuals in Petri dishes for 1 h (per species *n* = 5 to 6). As control, larvae were provided with solvent-treated host leaf discs (per species *n* = 3). The solvent was allowed to fully evaporate, before the treated leaves were offered to the larvae. After 1 h, leaf discs were mostly consumed and larvae were immediately frozen at -80˚C for later sample preparation.

## Preparation of insect samples

Insect samples were lyophilised and weighed and either extracted entirely (larvae of both species and adult *A. rosae*) or separated into thorax and abdomen (adult *A. rosae*). For a third group of *A. rosae* adults, surface washings were performed to test whether the focal substances are mostly distributed on the surface or occur in the insect body itself ([Fig 1]). Therefore, frozen C+ individuals were individually first vortexed for either 30 sec or 2 min in 500 µl ethyl acetate and the surface washes kept for later analyses (each *n* = 5). Washed individuals were transferred into new tubes and freeze-dried.

All freeze-dried body samples were homogenised and extracted in 130 µl of 100% ethyl acetate using an ultrasonic bath. Following that, body extracts and surface washes were dried under reduced pressure at 30˚C and resolved in 100 µl of methanol with 2.4 mg/l mefenemic acid (Sigma-Aldrich GmbH, Taufkirchen, Germany) as internal standard. All samples were filtered (polytetrafluoroethylene membrane, 0.2 µm pore size, Phenomenex, Torrance, CA, USA) and afterwards transferred to glass vials with inserts for measuring with the LC-MS.

## Metabolomics analysis and processing of data

Extracts were measured using an UHPLC (Dionex UltiMate 3000, Thermo Fisher Scientific, San José, CA, USA) equipped with a Kinetex XB-C18 column (1.7 µm, 150 × 2.1 mm, with

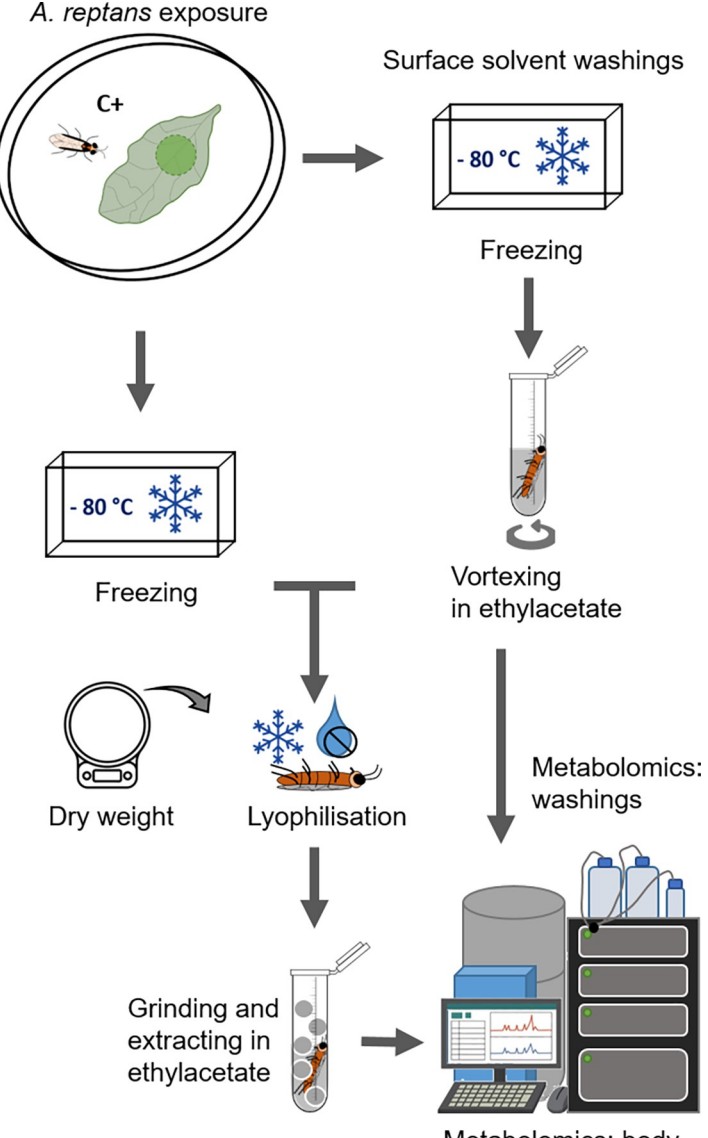

**Fig 1. Sample preparation of whole body samples and surface washings of adults of the sawfly *Athalia rosae*.**
Individuals exposed for 1 h to *Ajuga reptans* leaf discs (C+) or control individuals without contact (C-) were frozen.
Afterwards they were lyophilised, weighed and extracted or the surface was washed first and the washed body then
lyophilised and extracted. All samples were analysed by UHPLC-QToF-MS/MS.

guard column, Phenomenex) and coupled to a QToF-MS/MS (compact, Bruker Daltonics,
Bremen, Germany) in negative electrospray ionisation mode. The injection volume was 5 μl.
The column was preheated at 45°C and a solvent gradient from 0.1% formic acid (p.a., eluent
additive for LC-MS, ~98%, Sigma-Aldrich) in millipore water to 0.1% formic acid in acetoni-
trile (LC-MS grade, Fisher Scientific, Loughborough, UK; eluent B) was applied at a flow rate
of 0.5 ml min$^{-1}$. The proportion of eluent B started at 2% and increased to 30% B within 20
min, then to 75% B within 9 min. The ionisation was performed with a capillary voltage of
3000 V and an end plate offset of 500 V, while the nebuliser ($N_2$) pressure was 3 bar. The dry
gas ($N_2$) flow and temperature were 12 L min$^{-1}$ at 275°C. Line spectra were captured at 6 Hz,
with a range of 50–1300 *m/z*. For internal calibration, a Na(HCOO)-solution was constantly

injected to the source at the beginning of each sample to recalibrate the mass axis. Mass spectrometry was performed with an ion energy of 4 eV and the collision cell was set on collision energy of 7 eV, with a transfer time of 75 μs and pre-pulse storage of 6 μs.

Data was pre-checked using Compass DataAnalysis 4.4 (Bruker Daltonics, Bremen, Germany) and then processed via Metaboscape 2021b (Bruker Daltonics). Mass axis recalibration and peak picking with the T-ReX 3D algorithm including spectral background subtraction were performed in Metaboscape. The algorithm used an intensity threshold of 1000 counts and minimum peak length of 9 spectra. The allowed ion types for bucket generation were: [M-H]⁻, [M-H2O-H]⁻, [M+Cl]⁻, [M+HCOOH-H]⁻, [M+CH₃COOH-H]⁻ and [2M-H]⁻. Data was aligned in a bucket table with the same T-ReX 3D algorithm using the highest intensity as bucketing basis. The generated bucket table was then exported to Excel (2016, Microsoft, Washington, USA). For semi-quantification, intensities of the masses 484 and 482 were related to the intensities of mefenemic acid. The fragmentation patterns of these focal peaks were used to propose the chemical structure with competitive fragmentation modelling for metabolite identification (CFM-ID) [27, 28]. The data are available on Metabolights [29] with the identifier MTBLS7508.

## Statistical analysis

Data was analysed and visualised using R-Version 4.1.3 [30] and the package ggplot2 [31]. Metabolite concentrations in different body parts were statistically compared with Wilcoxon signed rank tests.

## Ethics

All animal experimentation was carried out in accordance to German legal and ethical standards. ARRIVE guidelines 2.0 were considered.

## Results and discussion

### Characterisation of clerodanoid-derived metabolites

Adults of *A. rosae* were either kept naïve, i.e. without contact to *A. reptans* (C-), or with contact to leaves or leaf extracts of *A. reptans* (C+). Comparisons of chromatograms from samples of C- and C+ adults of *A. rosae* and *A. reptans* leaf extracts revealed two focal peaks which were specific to C+ samples but did neither occur in C- samples nor in leaves. Thus, plant metabolites taken up from *A. reptans* are likely further metabolised in adults after nibbling on the leaves. The two peaks eluted at 16.3 min (*m/z* 529.229 [M+HCOOH-H]⁻) and 18.2 min (*m/z* 527.214 [M+HCOOH-H]⁻), and were resulting from the monoisotopic masses of 484.231 and 482.214 Da. Evaluation with Metaboscape resulted in the proposed sum formulas $C_{24}H_{36}O_{10}$ and $C_{24}H_{34}O_{10}$, respectively. The ratio of the two compounds was on average 2:3 throughout all samples of C+ individuals.

To further characterise the two focal compounds, their fragmentation patterns were assessed (Fig 2), using competitive fragmentation modelling for metabolite identification (CFM-ID) [27, 28], which is able to generate potential chemical structures. The molecular ion peak of compound 482 was observed at *m/z* 481.214, and its MS/MS spectrum displayed a series of fragment ions with *m/z* values of 463.197, 439.203, and 397.184, among others. For several fragment ions, chemical structures could be proposed, being consistent with the loss of various functional groups, including water ($H_2O$, 18 Da), formaldehyde ($CH_2O$, 30 Da) and ketene ($CH_2CO$, 42 Da). The first peak was connected to a chemical structure matching to that of the clerodanoid fatimanol A, as suggested by Metaboscape. Fatimanol A has been found in

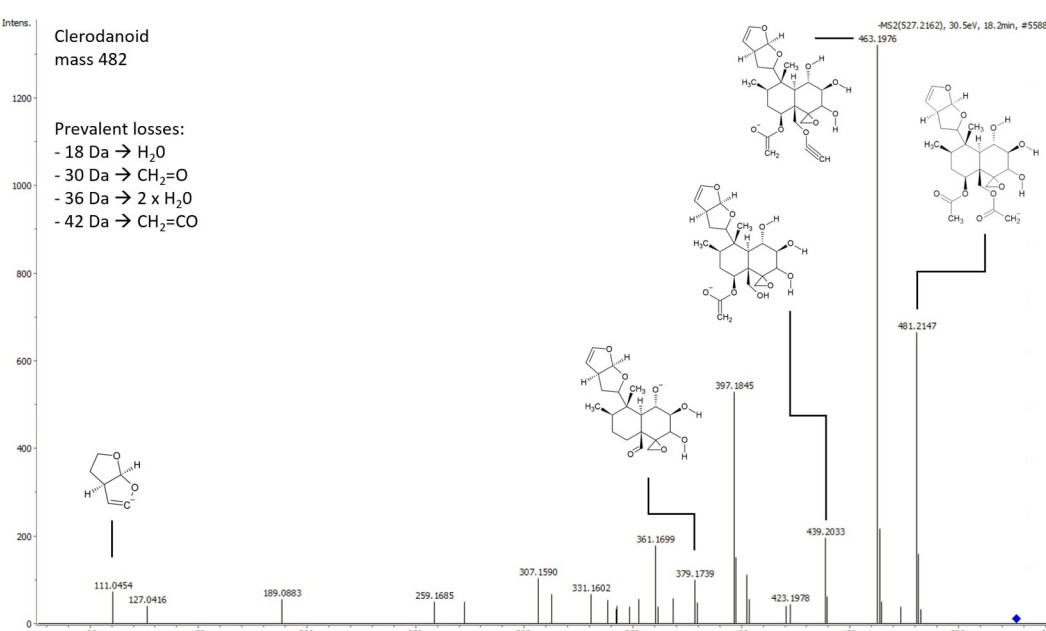

**Fig 2. MSMS-fragment spectra of compound 482.** The compound eluted at 18.2 min and was present in adult *Athalia rosae* after contact with *Ajuga reptans* leaves, measured with UHPLC-QToF-MS/MS at 6 Hz and 30.5 eV. Proposed chemical structures were generated using CFM-ID and matched to fragment peaks.

the Lamiaceae *Teucrium yemense* [32], which belongs to the subfamily of Ajugoideae, to which also *A. reptans* belongs. The base structure of fatimanol A has similarities to those of areptin A and B known from *A. reptans* [21], indicating that compound 482 found in *A. rosae* is also likely a clerodanoid. However, fatimanol A has a molecular weight of 482.5, which is not matching to our measured values. Thus, the compound detected in *A. rosae* cannot be fatimanol A. The fragment spectra of 484 showed a similar pattern in small *m/z* ranges, but no structure was proposed by Metaboscape. Since the sum formulas of $C_{24}H_{36}O_{10}$ and $C_{24}H_{34}O_{10}$ only differ in two hydrogen molecules, a similarity between the two compounds can be proposed. For further identification, nuclear magnetic resonance spectroscopy may be needed. However, generating sufficient material from these small insects is challenging, as the adults weigh less than 20 mg. If the extraction and purification of the focal metabolites in higher amounts would be possible, further bioassays may allow for testing of other biological activities apart from the already known functions in mating and defence [16, 18, 19].

The two focal compounds may be metabolised from the two clerodanoids areptin A and B that occur in *A. reptans* [21] (Fig 3). However, we were not able to detect the two areptins in leaf material, which may require further method optimisation. The two related metabolites detected in *A. rosae* adults show the bicyclic structure with the tetrahydrofurofuran ring and the epoxy group, characteristic for areptin A and B. These structures only differ in a double bond. We propose that either both clerodanoids are precursors for our two focal compounds or that only one of them acts as a precursor and the other one might be metabolised from the other metabolic product (Fig 3). Tetrahydrofurofuran might be the mediator of the biological activity. After exposure of adults of *A. rosae ruficornis* to *Clerodendrum trichotomum*, the two metabolites ajugachin A and athaliadiol could be isolated from body extracts [33]. Both of these compounds are clerodanoid-derived metabolites and share many structural similarities with the compounds detected in the present study in *A. rosae*.

**Fig 3. Proposed metabolism of clerodanoids by adults of *Athalia rosae*.** Proposed precursors (left) are the clerodanoids areptin A and B from *Ajuga reptans*, modified into the compounds with masses 484.231 and 482.214, respectively. Orange arrows indicate proposed metabolic changes.

## Focal metabolites occur in various body parts and on the body surface

Analysis of separated body parts and washings from individuals shed light on the distribution of metabolites in the body of adult *A. rosae* (Fig 4). In the thorax almost twice as high intensities were found than in the abdomen, differing significantly (Wilcoxon test, for both metabolites $V = 27$, $p = 0.022$), whereas whole individuals showed intensities almost as high as the sum of the amounts found in thorax and abdomen. Both compounds 482 and 484, showed a similar distribution pattern in the body parts.

Both focal compounds were not only detected in body parts, but also in small intensities in short (30 s and 2 min) surface washings in ethyl acetate (for 2 min: Fig 5). These results indicate that the compounds were indeed present on the surface. The washed bodies contained the substantial amounts of both compounds, highlighting that the compounds are incorporated into the body tissue. Nevertheless, findings of both focal compounds in surface washes support the idea that the adults distribute the two focal compounds via cleaning behaviour on their body or even may excrete them via potential glands as known for other species [34]. For example, adults of the pine sawfly, *Diprion similis* (Hymenoptera: Diprionidae), excrete pheromones from abdominal glands, followed by cleaning behavior and probably dispersion of these compounds over the body [35]. Adult *A. rosae* seem to metabolise and distribute the compounds within a short time span, as less than 30 min of contact with the leaves were sufficient to detect the metabolites afterwards in surface washings.

The presence of the two compounds in surface washings and thus on the surface of the adults can explain, how other adults are able to acquire the relevant substances via nibbling on conspecifics [17]. This interaction between conspecifics can lead to fights over the chemicals and thereby disrupt the mating behaviour. The processes of either uptake from leaves or acquisition of these compounds from conspecifics thus influence each individual's ecological as well as social niche [20, 36]. Interestingly, the amounts detected in body parts and on the surface

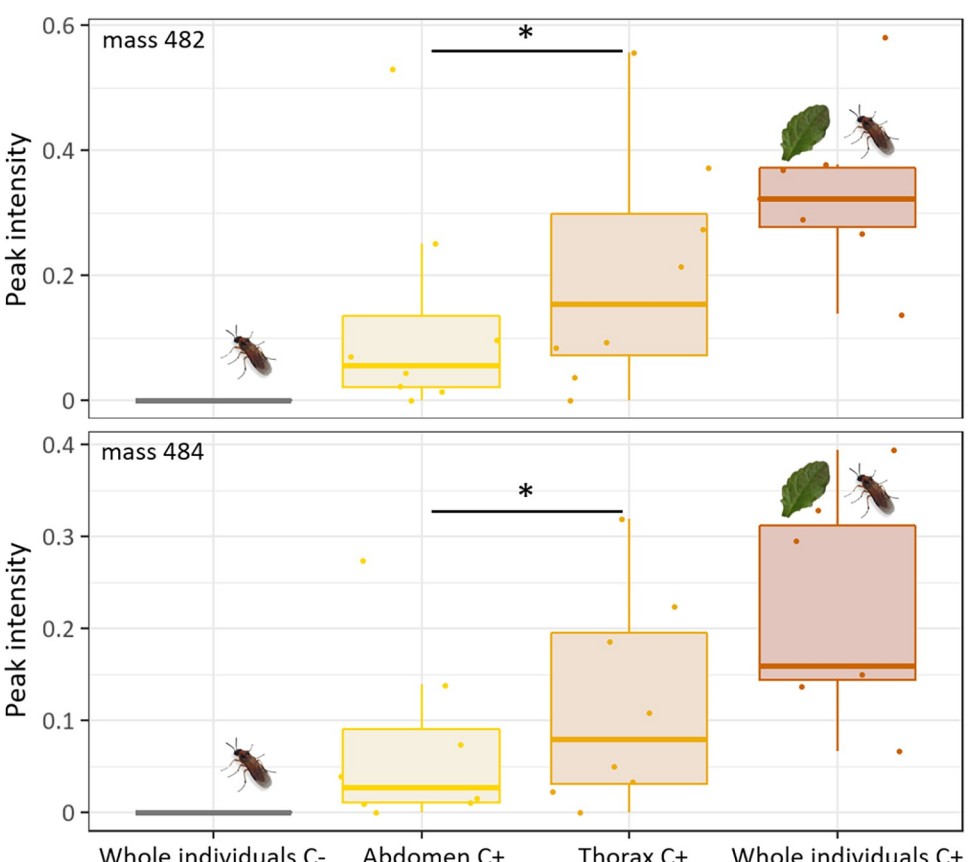

**Fig 4. Intensities of focal compounds in non-exposed individuals (C-), separated body parts (abdomen and thorax) and whole bodies of *Athalia rosae* after exposure to *Ajuga reptans* leaves (C+).** Intensities were related to the intensity of the internal standard. Box-plots show median, interquartile ranges (IQR) and whiskers indicate largest and smallest value up to 1.5 * IQR. Significant differences were tested between intensities found in thorax and abdomen (for both compounds Wilcoxon signed-rank test, $V = 27$, $p = 0.022$, indicated by the asterisk, $n = 6–8$).

varied quite substantially among the individuals, although all individuals had the same exposure time to the leaf material. Thus, individuals may have either nibbled for different durations within the exposure time on the leaves or some sawflies are more effective than others in acquiring and metabolising the pharmacological relevant substances. The two processes of acquiring metabolites from plants but also from conspecifics [17] furthermore depend on the availability of the plant in the insect populations and potentially on competition. Individual variation in sequestration of plant metabolites also occurs in other herbivorous species, such as pyrrolizidine alkaloid-imbibing caterpillars [37]. For other plant-acquired defences such as cantharidin, it is known that certain thresholds and concentrations limit their effect on predators [38]. Whether such limits also exist in *A. rosae* is currently unknown.

### Specificity of clerodanoid metabolism: Larvae of *A. rosae*, but not of *S. exigua*, are capable of metabolising focal compounds

Larvae of *A. rosae* and *S. exigua* were fed either with untreated cabbage leaves (control) or cabbage leaves spiked with *A. reptans* extract to test the specificity of metabolism of the clerodanoids. None of the samples from the control groups had detectable amounts of the focal compounds 484 and 482 (Fig 6). In *A. rosae* larvae exposed to *A. reptans* extracts both

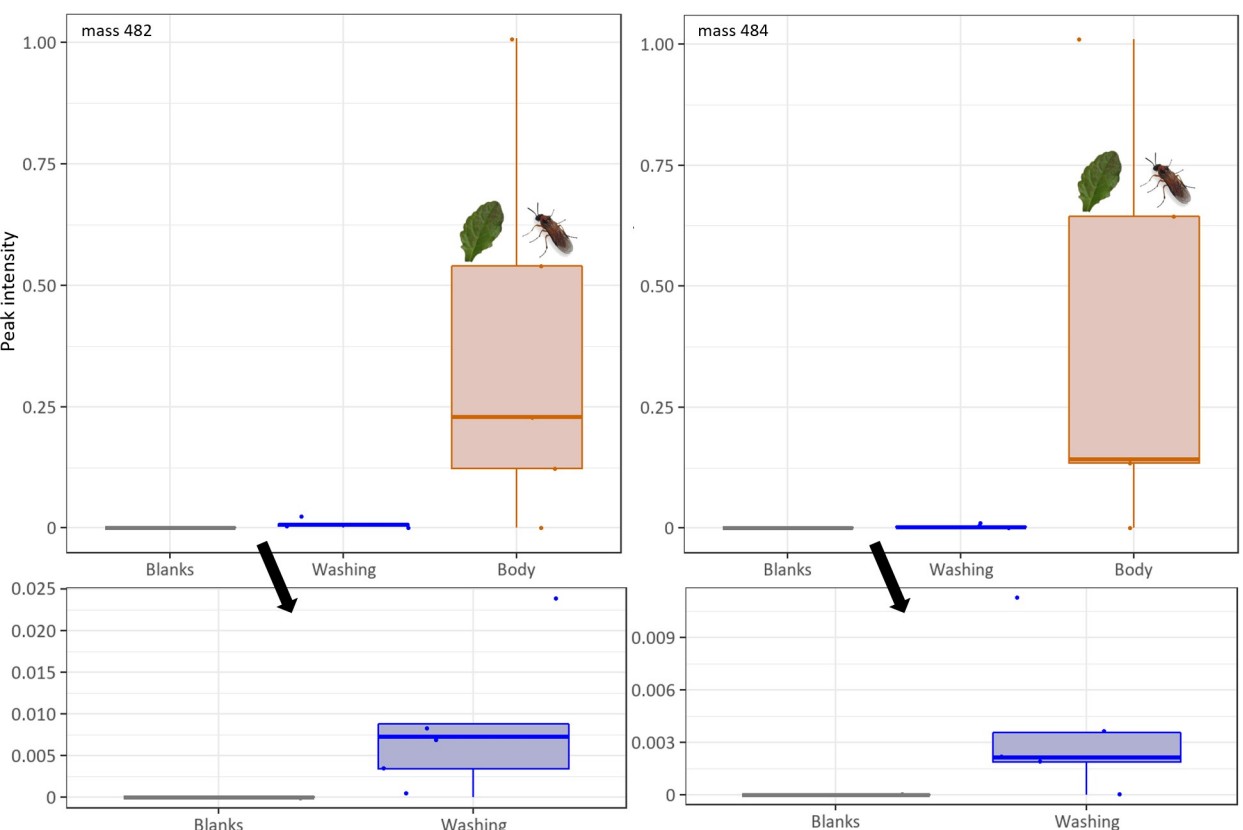

**Fig 5. Intensities of focal compounds 482 and 484 in surface washings and washed bodies of *Athalia rosae* after exposure to *Ajuga reptans* leaves (*n* = 5).** Intensities were related to the intensity of the internal standard. Box-plots show median, IQR and whiskers indicate largest and smallest value up to 1.5 * IQR.

compounds could be detected, however, in around ten times lower amounts compared to adult conspecifics. Thus, the larvae are already capable of metabolising these compounds, although they do not accept *A. reptans* leaves for feeding, as repeated trials in our laboratory revealed. The lower amounts detected in the larvae may be either due to a less effective metabolism or to the fact that amounts of the plant metabolites taken up differed between leaves (fed to the adults) and extracts fed to the larvae. The fact that the larvae of *A. rosae* are able to metabolise the clerodanoids although they do not feed in nature on *A. reptans* fits well to the hypothesis, that the genus *Athalia* may have originally used Lamiales as host plants but with radiation various species specialised as larvae on Brassicaceae [39]. Instead of acquiring compounds from *A. reptans*, *A. rosae* larvae rely in nature on defence through glucosinolate sequestration acquired from their Brassicaceae hosts, which provides benefits against predators [12, 13]. For future experiments it may be interesting to test whether individuals of *A. rosae* can keep the clerodanoid-compounds during metamorphosis into adulthood, as shown for glucosinolates [40]. Considering the phenomenon of pharmacophagy from the perspective of evolution, the behaviour may have evolved early in the phylogeny of several insect families and facultative pharmacophagous behaviour by larvae may be the derived condition. Even between subspecies the occurrence of pharmacophagy can differ, with either only adults, only larvae or both being able to take up plant specialised metabolites that are beneficial for the individuals [41].

In larvae of *S. exigua* fed with cabbage spiked with *A. reptans*, none of the two focal compounds could be detected (Fig 7). These findings show that the metabolism of the plant

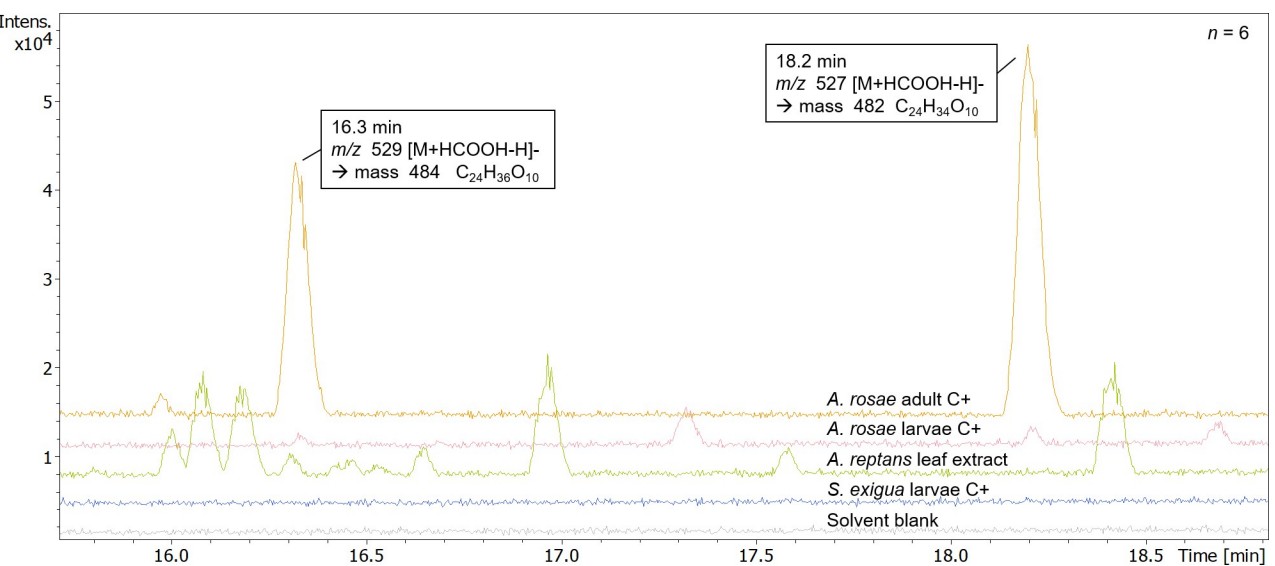

**Fig 6. Example chromatograms (UHPLC-QToF-MS/MS) of extracts from *Athalia rosae* individuals (adults and larvae), *Spodoptera exigua* larvae and *Ajuga reptans* leaf extracts in comparison, measured at 6 Hz.** Prior to extraction, insects were kept naive (C-, not shown here); or had access to a leaf (extract) of *Ajuga reptans* and thus to clerodanoids (C+). Chromatograms are zoomed in on the timeframe between 16 min and 18.5 min where focal peaks appear. Peaks visible in *A. rosae* are putative clerodanoids, which appear only after exposure to *A. reptans*: clerodanoid 482 ($C_{24}H_{34}O_{10}$) and candidate clerodanoid 484 ($C_{24}H_{36}O_{10}$). Peaks did not appear in C- individuals, *S. exigua* larvae or leaf extracts.

clerodanoids is specific to *A. rosae* or potentially also other closely related species, which may be based on specific enzymes. Since the focal compounds were not found in *S. exigua*, they may not just be modified due to our extraction procedure or, less likely, *S. exigua* may metabolise them even further.

Specialised plant metabolites clearly influence both resource usage as well as the foraging behaviour of organisms. Changes in foraging behaviour can result in an effective use of

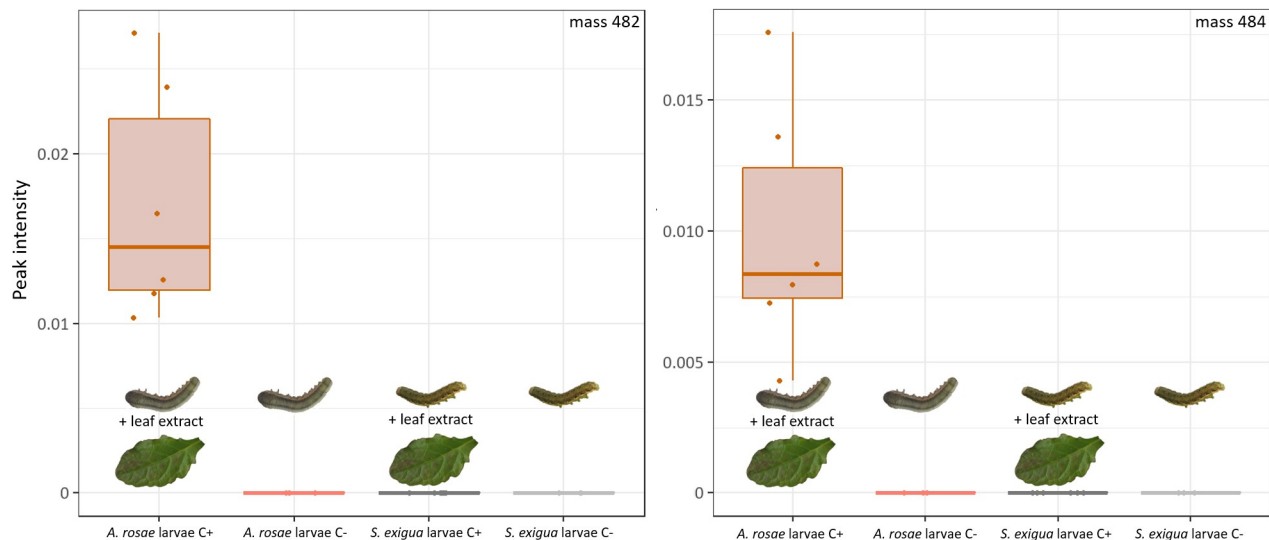

**Fig 7. Intensities of focal compounds in larvae of *A. rosae* and *S. exigua* after feeding on cabbage leaves without extract (C-) or spiked with *A. reptans* leaf extract (C+) (*n* = 6).** Intensities were related to the intensity of the internal standard. Box-plots show median, IQR and whiskers indicate largest and smallest value up to 1.5 * IQR.

defence compounds and an increased survival of populations [18]. In the case of *A. rosae*, larvae may use glucosinolates of their Brassicaceae host plants as feeding stimulants and certainly as defences, while adults likely use certain metabolites of Apiaceae nectar as feeding stimulants. In addition, the presence of clerodanoids in Lamiaceae may affect the decision-making process in the adults. Overall, specialised plant metabolites such as clerodanoids have the potential to influence the foraging behaviour and insect-plant interactions in ecosystems.

## Conclusion

In summary, our study suggests that two compounds from *A. reptans*, likely the clerodanoids areptin A and B, are further metabolised by adults of *A. rosae*. We propose chemical structures for the two compounds with the masses 482 and 484 ($C_{24}H_{34}O_{10}$ and $C_{24}H_{36}O_{10}$), which are likely also clerodanoids. These compounds are not only incorporated into the body but also brought to the surface via a yet unknown mechanism. Larvae are already able to metabolise these compounds, although naturally not coming into contact with them, pointing potentially to an ancient link to Lamiales in the *Athalia* genus. The individual variation in pharmacophagous uptake is likely affecting the ecological and social niche conformance in *A. rosae*. Our results shed light on the complex phenomenon of pharmacophagy in this sawfly species and its chemical peculiarities.

## Acknowledgments

We thank Sarah Catherine Paul for participating in very early stages of this research and providing some of the samples. We thank Alina Katharina Wessels for conducting some pilot experiments. Additionally, we thank the gardeners of Bielefeld University for rearing cabbage and bugleweed plants. Furthermore, thanks to Anke Stepphuhn and Sarah Isabel Richards for providing individuals of *S. exigua* and giving tips on handling and rearing. We acknowledge support for the publication costs by the Open Access Publication Fund of Bielefeld University and the Deutsche Forschungsgemeinschaft (DFG).

## Author Contributions

**Conceptualization:** Leon Brueggemann, Lisa Johanna Tewes, Caroline Müller.

**Data curation:** Leon Brueggemann, Caroline Müller.

**Formal analysis:** Leon Brueggemann.

**Investigation:** Leon Brueggemann, Lisa Johanna Tewes.

**Methodology:** Leon Brueggemann, Lisa Johanna Tewes.

**Project administration:** Caroline Müller.

**Resources:** Caroline Müller.

**Software:** Leon Brueggemann.

**Validation:** Leon Brueggemann, Caroline Müller.

**Visualization:** Leon Brueggemann.

**Writing – original draft:** Leon Brueggemann.

**Writing – review & editing:** Lisa Johanna Tewes, Caroline Müller.

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
