## [Decision Letter · Decision Letter 0]

1 Aug 2023

PONE-D-23-18631Characterisation and localisation of plant metabolites involved in pharmacophagy in the turnip sawflyPLOS ONE

Dear Dr. Müller,

Thank you for submitting your manuscript to PLOS ONE. After careful consideration, we feel that it has merit but does not fully meet PLOS ONE’s publication criteria as it currently stands. Therefore, we invite you to submit a revised version of the manuscript that addresses the points raised during the review process.

We look forward to receiving your revised manuscript.

Kind regards,

Mozaniel Santana de Oliveira, Ph.D

Academic Editor

PLOS ONE

“This research was funded by the German Research Foundation (DFG) as part of the SFB TRR 212 (NC³), project no. 396777467 (granted to C.M.)”

Reviewers' comments:

Reviewer's Responses to Questions

**Comments to the Author**

1. Is the manuscript technically sound, and do the data support the conclusions?

Reviewer #1: Yes

Reviewer #2: Yes

2. Has the statistical analysis been performed appropriately and rigorously? 

Reviewer #1: Yes

Reviewer #2: Yes

3. Have the authors made all data underlying the findings in their manuscript fully available?

Reviewer #1: Yes

Reviewer #2: Yes

4. Is the manuscript presented in an intelligible fashion and written in standard English?

Reviewer #1: Yes

Reviewer #2: Yes

5. Review Comments to the Author

Reviewer #1: 1. different timepoints of sample collection for metabolomic analysis can be done in the future experiment.

2. the potential function of focal metabolites will be very interesting to explore through bioassay.

3. a transcriptional comparison between C+ and C- adults may be helpful for mining the potential gene involved in the compound modification process.

Reviewer #2: The article is well written, but I recommend some adjustments for acceptance:

1st - I recommend reformulating the last paragraph of the introduction (lines 72 to 87). The way it was written, it fits better in materials and methods. I suggest a shorter paragraph presenting the research objectives.

2nd - all molecules eliminated in figure 3 have an extra carbon at the end of the broken bond. Fix it.

6. PLOS authors have the option to publish the peer review history of their article (what does this mean?). If published, this will include your full peer review and any attached files.

Reviewer #1: No

Reviewer #2: No

---

## [Author Response · Author response to Decision Letter 0]

14 Aug 2023

Dear Dr. Oliveira, 

Thank you very much for giving us the opportunity to revise our manuscript and thanks to all reviewers for their helpful comments. We carefully revised our manuscript following the suggestions by the reviewers and highlighted all changes. 

Please find below the comments by the handling editor and reviewers and our replies to each comment.

Kind regards,

Caroline Müller

GENERAL CHANGES

All references in the text were formatted within the sentences. instead of after the sentences. The reference for Metabolights was added (lines 175-176) and the ethics statement was moved up (lines 183-185). The following corrections in the figures were made:

Fig. 2: The rightmost structure was missing one negative load at the lowest CH2 group, which was added now.

Fig. 3: As correct stated by reviewer#2, the fragments in the middle section had redundant carbons and were removed.

Fig. 4: The label of the x-axis was changed to lower case letters.

Fig. 7: The label of the x-axis was changed to lower case letters.

All figures were checked by the PACE browser software.

COMMENTS BY REVIEWER #1 AND REPLIES

REVIEWER #1: 

1. different timepoints of sample collection for metabolomic analysis can be done in the future experiment.

REPLY: Thank you very much for the inspiring ideas for future projects. 

Different time points were sampled in the past, but did not really show a clear pattern. We would like to try to disentangle the effects in the future.

2. the potential function of focal metabolites will be very interesting to explore through bioassay.

REPLY: Yes, indeed, there may be even more effects than impacts on mating behaviour and defence. However, for bioassays these compounds would need to be purified, as they are not commercially available. We added a part about possible future bioassays in the manuscript in lines 218-221.

3. a transcriptional comparison between C+ and C- adults may be helpful for mining the potential gene involved in the compound modification process.

REPLY: In a previous study we investigated indeed differences in transcriptional pattern between C+ and C- adults, which can be found in Paul SC, Dennis AB, Tewes LJ, Friedrichs J, Müller C (2021) Consequences of pharmacophagous uptake from plants and conspecifics in a sawfly elucidated using chemical and molecular techniques. bioRxiv

https://doi.org/10.1101/2021.02.09.430406. As so little is known about the metabolism of these compounds, no clear suggestions about underlying genes can be made from this dataset yet.

COMMENTS BY REVIEWER #2 AND REPLIES

REVIEWER #2: The article is well written, but I recommend some adjustments for acceptance:

1. I recommend reformulating the last paragraph of the introduction (lines 72 to 87). The way it was written, it fits better in materials and methods. I suggest a shorter paragraph presenting the research objectives.

REPLY: Many thanks for your kind words and your helpful suggestions. We shortened the last paragraph of the introduction, highlighting the research objectives (now lines 72-84). Some parts were moved to the material and methods section (now lines 106ff).

2. All molecules eliminated in figure 3 have an extra carbon at the end of the broken bond. Fix it.

REPLY: Thank you for this important comment. We corrected the respective figure.

---

## [Decision Letter · Decision Letter 1]

23 Aug 2023

Characterisation and localisation of plant metabolites involved in pharmacophagy in the turnip sawfly

PONE-D-23-18631R1

Dear Dr. Müller,

We’re pleased to inform you that your manuscript has been judged scientifically suitable for publication and will be formally accepted for publication once it meets all outstanding technical requirements.

Kind regards,

Mozaniel Santana de Oliveira, Ph.D

Academic Editor

PLOS ONE

Additional Editor Comments (optional):

Reviewers' comments:

Reviewer's Responses to Questions

**Comments to the Author**

1. If the authors have adequately addressed your comments raised in a previous round of review and you feel that this manuscript is now acceptable for publication, you may indicate that here to bypass the “Comments to the Author” section, enter your conflict of interest statement in the “Confidential to Editor” section, and submit your "Accept" recommendation.

Reviewer #1: All comments have been addressed

Reviewer #2: All comments have been addressed

2. Is the manuscript technically sound, and do the data support the conclusions?

Reviewer #1: Yes

Reviewer #2: Yes

3. Has the statistical analysis been performed appropriately and rigorously? 

Reviewer #1: Yes

Reviewer #2: Yes

4. Have the authors made all data underlying the findings in their manuscript fully available?

Reviewer #1: Yes

Reviewer #2: Yes

5. Is the manuscript presented in an intelligible fashion and written in standard English?

Reviewer #1: Yes

Reviewer #2: Yes

6. Review Comments to the Author

Reviewer #1: The authors have satisfactorily responded to all my questions and made the necessary changes to the manuscript. Therefore, I have no further comments.

Reviewer #2: The authors complied with all suggested recommendations and the article is ready for publication. Therefore, I recommend acceptance of the manuscript.

7. PLOS authors have the option to publish the peer review history of their article (what does this mean?). If published, this will include your full peer review and any attached files.

Reviewer #1: No

Reviewer #2: No

---

## [Editor Report · Acceptance letter]

28 Sep 2023

PONE-D-23-18631R1 

Characterisation and localisation of plant metabolites involved in pharmacophagy in the turnip sawfly 

Dear Dr. Müller:

I'm pleased to inform you that your manuscript has been deemed suitable for publication in PLOS ONE. Congratulations! Your manuscript is now with our production department. 

Kind regards, 

on behalf of

Dr. Mozaniel Santana de Oliveira 

Academic Editor

PLOS ONE